# Understanding the Genetic Landscape of Pancreatic Ductal Adenocarcinoma to Support Personalized Medicine: A Systematic Review

**DOI:** 10.3390/cancers16010056

**Published:** 2023-12-21

**Authors:** Antonino Pantaleo, Giovanna Forte, Candida Fasano, Martina Lepore Signorile, Paola Sanese, Katia De Marco, Elisabetta Di Nicola, Marialaura Latrofa, Valentina Grossi, Vittoria Disciglio, Cristiano Simone

**Affiliations:** 1Medical Genetics, National Institute of Gastroenterology-IRCCS “Saverio de Bellis” Research Hospital, 70013 Bari, Italy; pantaleo.labsimone@gmail.com (A.P.); forte.labsimone@gmail.com (G.F.); fasano.labsimone@gmail.com (C.F.); leporesignorile.labsimone@gmail.com (M.L.S.); sanese.labsimone@gmail.com (P.S.); demarco.labsimone@gmail.com (K.D.M.); dinicola.labsimone@gmail.com (E.D.N.); m.latrofa1995@outlook.it (M.L.); grossi.labsimone@gmail.com (V.G.); 2Medical Genetics, Department of Precision and Regenerative Medicine and Jonic Area (DiMePRe-J), University of Bari Aldo Moro, 70124 Bari, Italy

**Keywords:** pancreatic adenocarcinoma, personalized medicine, germline variant, genetic risk assessment

## Abstract

**Simple Summary:**

Pancreatic ductal adenocarcinoma (PDAC) is an aggressive cancer with high mortality. Most patients present with an advanced stage of the disease, highlighting the urgent need for early detection. Recent studies of individuals at high risk of PDAC showed benefits from participating in clinical management and surveillance programs. PDAC clinical management and surveillance programs are suggested for individuals with a germline pathogenic variant in a cancer predisposition gene or a strong family history. In the present study, we performed a systematic literature review to investigate the mutational portrait of the main genes (*ATM*, *BRCA1*, *BRCA2*, *CDKN2A*, *EPCAM*, *MLH1*, *MSH2*, *MSH6*, *PALB2*, *PMS2*, *STK11*, *TP53*) involved in PDAC susceptibility. Our findings may support the development of tailored management and follow-up strategies in PDAC patients with specific germline genetic variants.

**Abstract:**

Pancreatic ductal adenocarcinoma (PDAC) is one of the most fatal malignancies worldwide. While population-wide screening recommendations for PDAC in asymptomatic individuals are not achievable due to its relatively low incidence, pancreatic cancer surveillance programs are recommended for patients with germline causative variants in PDAC susceptibility genes or a strong family history. In this study, we sought to determine the prevalence and significance of germline alterations in major genes (*ATM*, *BRCA1*, *BRCA2*, *CDKN2A*, *EPCAM*, *MLH1*, *MSH2*, *MSH6*, *PALB2*, *PMS2*, *STK11*, *TP53*) involved in PDAC susceptibility. We performed a systematic review of PubMed publications reporting germline variants identified in these genes in PDAC patients. Overall, the retrieved articles included 1493 PDAC patients. A high proportion of these patients (*n* = 1225/1493, 82%) were found to harbor alterations in genes (*ATM*, *BRCA1*, *BRCA2*, *PALB2*) involved in the homologous recombination repair (HRR) pathway. Specifically, the remaining PDAC patients were reported to carry alterations in genes playing a role in other cancer pathways (*CDKN2A*, *STK11*, *TP53*; *n* = 181/1493, 12.1%) or in the mismatch repair (MMR) pathway (*MLH1*, *MSH2*, *MSH6*, *PMS2*; *n* = 87/1493, 5.8%). Our findings highlight the importance of germline genetic characterization in PDAC patients for better personalized targeted therapies, clinical management, and surveillance.

## 1. Introduction

Pancreatic ductal adenocarcinoma (PDAC) is one of the most fatal malignancies worldwide, with a 5-year overall survival of about 5–10% and a 10-year overall survival of less than 1% [1,2]. Recent epidemiological data indicate that the incidence of PDAC is rising. It currently ranks 12th among the most common cancers worldwide and is the seventh-highest cause of cancer death in both sexes [3,4]. Most patients present with surgically unresectable disease or distant metastases, highlighting the urgent need for early detection [5]. The most common metastatic sites for PDAC are the liver and peritoneum. Less frequently, lung, brain, and bone metastases are also detected in PDAC patients [6]. Focusing on early diagnosis of metastatic PDAC, with a particular emphasis on unusual symptoms such as bone pain, which might be related to skeletal metastases, is a key priority to extend survival and improve the quality of life of PDAC patients [7].

Currently, population-wide screening recommendations for PDAC in asymptomatic individuals are not achievable due to the relatively low incidence of this cancer [8]. Conversely, pancreatic surveillance programs are recommended for patients with germline causative variants in genes involved in PDAC susceptibility or a strong family history [8]. Nowadays, it is estimated that approximately 10% of individuals with PDAC have alterations in genes associated with known hereditary cancer syndromes that are also associated with an increased risk of developing this disease [9]. More specifically, patients with pathogenic or likely pathogenic variants (PVs/LPVs) in the *BRCA1* and *BRCA2* genes, which are associated with hereditary breast and ovarian cancer, have a 5–10% increased risk of developing PDAC [10]. Likewise, a higher risk of developing PDAC has been associated with germline genetic alterations in the *MLH1*, *MSH2*, *MSH6*, and *EPCAM* genes, which are involved in Lynch syndrome (LS) [11], and in the *STK11* gene, which is responsible for Peutz–Jeghers syndrome [12]. Additionally, germline PVs/LPVs in other genes such as *PALB2*, *ATM*, *CDKN2A,* and *TP53*, which are involved in other hereditary cancer syndromes, also confer an increased risk of PDAC [8,13]. The identification of PVs/LPVs in one of these PDAC susceptibility genes has relevant implications for therapeutic strategies. For example, patients harboring genetic alterations in *BRCA1* or *BRCA2* may benefit from targeted therapies [14]. Considering the clinical impact of this evidence, the National Comprehensive Cancer Network (NCCN) currently recommends genetic testing for all individuals with a PDAC diagnosis [15]. In addition, the American Society of Clinical Oncology released a provisional clinical opinion according to which all individuals with newly diagnosed PDAC should undergo risk assessment for hereditary cancer syndromes known to be associated with an increased risk of PDAC, and germline genetic testing should be offered to patients with a personal history of pancreatic cancer, even if they have no remarkable family history [16]. In the present study, we performed a systematic literature review to investigate the mutational portrait of the main genes involved in PDAC susceptibility.

## 2. Materials and Methods

### 2.1. Search Strategy and Study Selection

This systematic review was conducted according to the PRISMA (Preferred Reporting Items for Systematic Reviews and Meta–Analyses) guidelines (Appendix A) [17] and was not registered in PROSPERO (international prospective register of systematic reviews). It was performed in PubMed by searching for studies reporting PDAC cases carrying germline variants in the main genes associated with PDAC susceptibility. The timeframe of the search was from February 1976 to September 2023. The search comprised the keywords (((pancreatic cancer) OR (pancreatic adenocarcinoma) OR (pancreatic ductal adenocarcinoma)) AND ((germline variant) OR (germline mutation) OR (pathogenic germline variant)) NOT (neuroendocrine)). Only studies including PDAC patients harboring a germline genetic variant in one of the following PDAC susceptibility genes were selected: *ATM*, *BRCA1*, *BRCA2*, *CDKN2A*, *EPCAM*, *MLH1*, *MSH2*, *MSH6*, *PALB2*, *PMS2*, *STK11*, and *TP53*. The population of interest was composed of individuals with any type or stage of PDAC. Studies reporting patients with pancreatic neuroendocrine tumors or PDAC patients not harboring disease-associated germline variants, or PDAC patients with only somatic alterations in PDAC susceptibility genes were excluded. Studies not including relevant information (genetic sequence variant nomenclature) were also excluded. Moreover, patients described in different studies were included only once.

### 2.2. Data Extraction

Ten reviewers independently assessed the title, abstract, main text, and Appendix A of the identified articles to determine study inclusion or exclusion. Relevant information regarding PDAC patients with disease-causative genetic sequence variants was extracted from the full text of the included articles. Any disagreement was resolved by a consensus meeting among the reviewers. The extracted study details included information about the article (author(s), year of publication, title, DOI), genetic sequence variant nomenclature (gene, gene transcript, DNA change, protein change, type of variant), and the number of patients carrying each genetic variant.

## 3. Results

### 3.1. Literature Search and Study Selection

The initial search of the literature using the keywords chosen based on the inclusion criteria retrieved 912 records. After screening these 912 records, no article was excluded. The full-text articles were studied and assessed for retrieval, and 66 studies were not retrieved and, therefore, excluded. The remaining 846 studies were assessed for eligibility. Among these, 527 studies were excluded for different reasons (articles not in English, PDAC patients without PVs/LPVs in the selected genes, patients carrying only somatic alterations in PDAC susceptibility genes), and 150 studies were excluded because the evaluated patients had endocrine pancreatic cancer. As a result, 169 published studies fulfilled the inclusion criteria [11,15,18,19,20,21,22,23,24,25,26,27,28,29,30,31,32,33,34,35,36,37,38,39,40,41,42,43,44,45,46,47,48,49,50,51,52,53,54,55,56,57,58,59,60,61,62,63,64,65,66,67,68,69,70,71,72,73,74,75,76,77,78,79,80,81,82,83,84,85,86,87,88,89,90,91,92,93,94,95,96,97,98,99,100,101,102,103,104,105,106,107,108,109,110,111,112,113,114,115,116,117,118,119,120,121,122,123,124,125,126,127,128,129,130,131,132,133,134,135,136,137,138,139,140,141,142,143,144,145,146,147,148,149,150,151,152,153,154,155,156,157,158,159,160,161,162,163,164,165,166,167,168,169,170,171,172,173,174,175,176,177,178,179,180,181,182,183,184]. The selected 169 published studies included different types of study designs, mostly consisting of case studies and cohort studies (Appendix A). The language of all articles was English. Figure 1 shows the flow diagram for study selection.

### 3.2. Gene Alteration Frequency in PDAC Patients

Overall, the identified articles included 1493 PDAC patients who were found to carry PVs/LPVs in one of the selected genes. Genetic testing was performed in PDAC individuals with or without a family history of different cancer types (i.e., pancreatic, breast, colorectal, or melanoma). Full details of all the genetic causative or likely causative variants reported are shown in Appendix A [11,15,18,19,20,21,22,23,24,25,26,27,28,29,30,31,32,33,34,35,36,37,38,39,40,41,42,43,44,45,46,47,48,49,50,51,52,53,54,55,56,57,58,59,60,61,62,63,64,65,66,67,68,69,70,71,72,73,74,75,76,77,78,79,80,81,82,83,84,85,86,87,88,89,90,91,92,93,94,95,96,97,98,99,100,101,102,103,104,105,106,107,108,109,110,111,112,113,114,115,116,117,118,119,120,121,122,123,124,125,126,127,128,129,130,131,132,133,134,135,136,137,138,139,140,141,142,143,144,145,146,147,148,149,150,151,152,153,154,155,156,157,158,159,160,161,162,163,164,165,166,167,168,169,170,171,172,173,174,175,176,177,178,179,180,181,182,183,184]. Overall, most of these PDAC patients (*n* = 1225, 82%) were found to harbor PVs/LPVs in genes (*ATM*, *BRCA1*, *BRCA2, PALB2*) involved in the DNA homologous recombination repair (HRR) pathway. Among this subset of patients, 561 (45.8%) carried PVs/LPVs in *BRCA2*, 310 (25.3%) in *ATM,* 239 (19.5%) in *BRCA1*, and 115 (9.4%) in *PALB2*. A lower proportion of patients (*n* = 181, 12.1%) were reported to harbor PVs/LPVs in genes (*CDKN2A*, *STK11*, *TP53*) involved in other cancer-related pathways. Of these 181 patients, 146 (80.7%) carried PVs/LPVs in *CDKN2A*, 24 (13.2%) in *TP53*, and 11 (6.1%) in *STK11*. The remaining 87 (5.8%) patients were found to harbor PVs/LPVs in genes (*MLH1*, *MSH2*, *MSH6*, and *PMS2*) involved in the DNA mismatch repair (MMR) pathway. Of these 87 patients, 26 (29.9%) carried PVs/LPVs in *MSH6*, 23 (26.4%) in *MLH1*, 20 (23%) in *PMS2,* and 18 (20.7%) in *MSH2* (Figure 2).

### 3.3. Types of Variants Identified in HRR Genes (ATM, BRCA1, BRCA2, PALB2)

As indicated above, a total of 1225 out of 1493 (82%) PDAC patients carried PVs/LPVs in genes involved in the HRR pathway (Figure 2 and Appendix A) [11,15,18,19,20,21,22,23,24,25,26,27,28,29,30,31,32,33,34,35,36,37,38,39,40,41,42,43,44,45,46,47,48,49,50,51,52,53,54,55,56,57,58,59,60,61,62,63,64,65,66,67,68,69,70,71,72,73,74,75,76,77,78,79,80,81,82,83,84,85,86,87,88,89,90,91,92,93,94,95,96,97,98,99,100,101,102,103,104,105,106,107,108,109,110,111,112,113,114,115,116,117,118,119,120,121,122,123,124,125,126,127,128,129,130,131,132,133,134,135,136,137,138,139,140,141,142,143,144,145,146,147,148,149,150,151,152,153,154,155,156,157,158,159,160,161,162,163,164,165,166,167,168,169,170,171,172,173,174,175,176,177,178,179,180,181,182,183,184].

Among the 561 patients with *BRCA2* germline PVs/LPVs (*n* = 308), 258 truncating variants (171 frameshift and 87 nonsense) were identified in 497 (88.5%) patients, 26 splicing alterations were identified in 32 (5.7%) patients, 20 missense variants were identified in 28 (5%) patients, and a unique in-frame deletion variant was identified in a single (0.2%) patient. Moreover, three (0.5%) patients were found to harbor different large deletions involving the *BRCA2* gene (Figure 3a and Appendix A).

Among the 239 patients with *BRCA1* germline PVs/LPVs (*n* = 95), 70 truncating variants (50 frameshift and 20 nonsense) were identified in 198 patients (82.8%), nine splicing variants were identified in 10 (4.2%) patients, and seven missense variants were identified in 20 patients (8.4%). Moreover, 11 (4.6%) patients were found to carry nine different copy number variations (CNVs) involving the *BRCA1* gene (Figure 3b and Appendix A).

Among the 310 patients with *ATM* germline PVs/LPVs (*n* = 198), 137 truncating variants (81 frameshift and 56 nonsense) were identified in 224 (72.2%) patients, 31 splicing variants were identified in 47 (15.2%) patients, 21 missense variants were identified in 25 patients (8.1%), and three in–frame variants were identified in seven (2.3%) patients. Moreover, six CNVs (large deletions) involving the *ATM* gene were found in seven (2.3%) patients (Figure 3c and Appendix A).

Among the 115 patients with *PALB2* germline PVs/LPVs (*n* = 70), 57 truncating variants (38 frameshift and 19 nonsense) were identified in 98 (85.2%) patients, five splicing variants were identified in eight (7%) patients, and four missense variants were identified in four (3.5%) patients. Additionally, four CNVs were identified in five (4.3%) patients (Figure 3d and Appendix A).

### 3.4. Types of Variants Identified in MMR Genes (EPCAM, MLH1, MSH2, MSH6, and PMS2)

As indicated above, a total of 87 out of 1493 (5.8%) PDAC patients carried PVs/LPVs in genes involved in the MMR (Figure 2 and Appendix A). No individual was found to harbor germline causative genetic alterations involving the *EPCAM* gene.

Among the 23 patients with *MLH1* germline PVs/LPVs (*n* = 19), five truncating variants (three nonsense and two frameshift) were identified in five patients (21.7%), four splicing variants were identified in seven patients (30.4%), and seven missense variants were identified in eight patients (34.8%). Additionally, two in-frame deletions were identified in two patients (8.7%). As regards CNVs, a unique CNV (large duplication involving *MLH1* exons 9 and 12) was found in one patient (4.4%) (Figure 4a and Appendix A).

Among the 18 patients with *MSH2* germline PVs/LPVs (*n* = 12), six truncating variants (five nonsense and one frameshift) were identified in six patients (33.4%), and a single splicing variant was identified in one patient (5.6%). Moreover, two missense variants were identified in five patients (27.8%). In addition, two patients (11.1%) were found to carry one in-frame deletion and four patients (22.2%) were found to carry two different CNVs (large *MSH2* deletions) (Figure 4b and Appendix A).

Among the 26 patients with *MSH6* germline PVs/LPVs (*n* = 25), 21 truncating variants (14 frameshift and seven nonsense) were identified in 22 patients (84.6%), two splicing variants were identified in two patients (7.7%), and a unique missense variant was identified in one patient (3.8%). Moreover, one patient (3.8%) was found to harbor a deletion involving *MSH6* exon 2 (Figure 4c and Appendix A).

Among the 20 patients with *PMS2* germline PVs/LPVs (*n* = 16), eight truncating variants (five frameshift and three nonsense) were identified in nine patients (45%), two splicing variants were identified in two patients (10%), and three missense variants were identified in six patients (30%). Additionally, three CNVs (large deletions) involving the *PMS2* gene were found in three patients (15%) (Figure 4d and Appendix A).

### 3.5. Types of Variants Identified in Other Cancer–Related Genes (CDKN2A, STK11, TP53)

As indicated above, a total of 181 out of 1493 (12.1%) PDAC patients carried PVs/LPVs in genes (*CDKN2A*, *STK11*, *TP53*) involved in other cancer-related pathways (Figure 2 and Appendix A).

Our analysis identified 146 patients with *CDKN2A* germline PVs/LPVs (*n* = 41). More specifically, 18 truncating variants (13 frameshift and five nonsense) were identified in 49 patients (33.6%), a single splicing variant was identified in three patients (2.1%), 19 missense variants were found in 90 patients (61.6%), and three in-frame deletions/duplications were identified in four patients (2.7%) (Figure 5a and Appendix A).

Among the 24 patients with *TP53* germline PVs/LPVs (*n* = 20), we detected five truncating variants (three frameshift and two nonsense) in six patients (25%), and 15 missense variants in 18 patients (75%) (Figure 5b and Appendix A).

Among the 11 patients with germline PVs/LPVs (*n* = 10) in the *STK11* gene, we detected seven truncating variants (two frameshift and five nonsense) in eight patients (72.4%), and a unique splicing variant in a single patient (9.1%). Moreover, two deletions involving the promoter and part of the coding region (exon 1 and exons 1–3) of the *STK11* gene were found in two patients (18.2%) (Figure 5d and Appendix A).

## 4. Discussion

In this study, we sought to determine the distribution and type of germline pathogenic variants in the main genes associated with PDAC susceptibility. The identification of germline PVs/LPVs in susceptibility genes may enable specific surveillance programs to be provided to individuals at high risk of developing PDAC [185]. Specific PDAC surveillance programs have been proven to support the identification of premalignant lesions or pancreatic cancer at an early stage in high–risk individuals with or without a family history of PDAC [186].

Based on the PDAC patients with germline PVs/LPVs identified through this systematic literature review, PVs/LPVs were most frequently reported in the *ATM*, *BRCA1*, *BRCA2*, and *PALB2* genes. All these genes participate in the HRR pathway, which repairs DNA double–strand breaks (DSBs) [112,187]. BRCA1 and BRCA2 play a pivotal role in DNA damage response via the HRR pathway [188]. BRCA1/2–deficient cells lack HRR activity and accumulate DSBs, which results in genomic instability and increased cancer risk [189]. Germline genetic alterations affecting *BRCA1*/*2* are primarily responsible for breast and ovarian cancer. In addition, PVs/LPVs involving these genes are associated with an increased risk of colon and prostate cancer [190]. Moreover, *BRCA1/2* germline alterations have been reported in about 5–10% of patients with familial PDAC and about 3% of patients with apparently sporadic PDAC [99,112]. In the current study, we determined the frequency and distribution of HRR gene alterations in the PDAC patients reported in the articles included in our literature review.

Overall, a higher rate of PDAC patients (561/1225, 45.8%) was found to harbor PVs/LPVs in the *BRCA2* gene as compared to PDAC patients harboring PVs/LPVs in other HRR genes (*BRCA1*, *ATM*, and *PALB2*). The vast majority of these patients (497/561, 88.5%) carried truncating variants, among which six alterations (c.3170_3174del, p.Lys1057fs; c.3847_3848del, p.Val1283Lysfs; c.5946del, p.Ser1982Argfs*22; c.6174del, p.Phe2058Leufs*12; c.6275_6276del, p.Leu2092Profs*7, c.8537_8538del, p.Glu2846Glyfs*22) recurred frequently.

In particular, the *BRCA2* c.5946del (p.Ser1982Argfs*) recurring variant is reported to be a founder mutation in the Ashkenazi Jewish population [191] and was identified in 56 PDAC patients. Other BRCA2 variant types (i.e., missense, in–frame deletion, CNV, and splicing) were identified with a low frequency in PDAC patients.

As regards PDAC patients (*n* = 239) harboring *BRCA1* germline alterations, most of them (198/239, 82.8%) had truncating variants. Of these, three were commonly occurring founder mutations (c.68_69delAG, p.Glu23fs; c.5266dup, p.Gln1756Pfs*74; c.5319dup, p.Asn1774Glnfs*56) [130,192,193]. Other *BRCA1* variant types (i.e., missense, and splicing) were identified with a lower frequency in PDAC patients (12.6%) as compared to truncating variants.

The initial phase of the HRR pathway is orchestrated by the kinases ATM and ATR, which are responsible for the phosphorylation of several proteins involved in downstream steps of the repair cascade [187]. Biallelic germline variants in the ATM gene cause ataxia–telangiectasia, a disorder characterized by neuronal degeneration, immune deficiency, and increased cancer risk [194]. On the other hand, monoallelic *ATM* germline variants have been shown to be associated with an increased risk of developing malignancies, including breast, pancreatic, prostate, stomach, ovarian, colorectal, and melanoma [195]. Importantly, monoallelic *ATM* germline alterations have also been reported in familial and sporadic cases of PDAC [67]. In addition, biallelic *ATM* inactivation has been found with a higher frequency in PDAC specimens from familial pancreatic cancer patients compared to sporadic cases [196]. In one study, the relative risk of pancreatic cancer was estimated to be 2.41 in patients carrying a monoallelic *ATM* germline variant [197]. In our literature review analysis, 198 distinct germline mutations in the *ATM* gene were detected in 310 (25.3%) PDAC patients. Within this subset, a high proportion of PVs/LPVs (137, 69.2%) were truncating variants. Four of these truncating variants (c.1564_1565del, p.Glu522fs; c.3245_3247delinsTGAT, p.His1082Leufs*14; c.3802del, p.Val1268fs; c.5932G>T, p.Glu1978*) recurred in more than five PDAC patients. Interestingly, among the several *ATM* splicing variants (*n* = 31, 15.6%) identified in 47 PDAC patients in our literature review, one (c.7630-2A>C) recurred in 10 patients.

The *PALB2* gene plays a crucial role in the HRR pathway via modulation of BRCA2 and RAD51 recruitment to DSBs. Germline *PALB2* pathogenic variants have been associated with an increased risk of breast, ovarian, and pancreatic cancer [167,198]. Our literature review analysis showed that a lower rate of PDAC patients (115/1225, 9.4%) was found to harbor PVs/LPVs in the *PALB2* gene as compared to PDAC patients harboring PVs/LPVs in other HRR genes (*BRCA1*, *BRCA2*, and *ATM*). In this subgroup, most patients (98/115, 85.2%) carried truncating variants. Four of these *PALB2* truncating variants (c.172_175del, p.Gln60Argfst*7; c.1240C>T, p.Arg414*; c.3116del, p.Asn1039fs; c.3256C>T, p.Arg1086*) recurred in five or more PDAC unrelated patients. Interestingly, other *PALB2* variant types (i.e., missense and splicing) were identified with a lower frequency in PDAC patients (*n* = 12, 10.5%) as compared to truncating variants. The remaining four (5.7%) genetic alterations were large *PALB2* deletions/duplications and were identified in five (4.3%) patients. Of note, all of them involved the C–terminal region of the gene.

Clinical oncology research is increasingly focusing on the identification of germline and somatic alterations in HRR genes as a way to guide targeted therapy in different types of cancers, including PDAC. Recent retrospective [85,199] and prospective studies [200] examining the use of the DSB–inducing drug cisplatin in PDAC patients with altered *BRCA1/2* have shown clinical benefit. Moreover, poly(ADP–ribose) polymerase (PARP) inhibitors have shown promise as a treatment for tumors with *BRCA* mutations. In particular, PARP inhibitors were found to increase progression–free survival when used as a maintenance therapy for patients with pancreatic cancer who responded to first–line platinum–based therapy [14].

Recent studies have shown that germline alterations in MMR genes (*MLH1*, *MSH2*, *MSH6*, *PMS2*, and *EPCAM*) are primarily responsible for LS, a common hereditary colorectal cancer syndrome [201,202]. LS confers an increased risk of multiple types of malignancies, including colorectal, endometrial, ovarian, stomach, small intestine, hepatobiliary system, urinary tract, and brain cancer [201,202]. The loss of DNA MMR activity determines the high microsatellite instability (MSI–H) phenotype observed in tumor samples of these patients [85,200]. Moreover, patients with a molecular diagnosis of LS also have an estimated 9– to 11–fold increased risk of pancreatic cancer [203,204].

To obtain further insight into the involvement of MMR gene alterations in PDAC, we determined their frequency and distribution in the PDAC patients reported in the articles included in our literature review. Collectively, germline mutations in the *MLH1*, *MSH2*, *PMS2*, and *MSH6* genes were detected in 5.8% of PDAC patients, with comparable frequencies for each of these genes. No germline alteration was identified in the *EPCAM* gene.

As regards the *MLH1* genetic alterations identified in PDAC patients, they were evenly distributed throughout the coding region, without obvious hotspots. Among these PVs/LPVs, a similar proportion of missense and truncating variants was observed in PDAC patients. Interestingly, three patients harbored the splicing variant c.677+3A>G, which has been reported to cause skipping of *MLH1* exon 8 [205]. Furthermore, two PDAC patients carried the splicing variant c.1731G>A (p.Ser577=), which is located next to the exon 15 splicing donor site and has been functionally shown to cause skipping of *MLH1* exon 15 [206]. Additionally, two patients were found to carry the missense variant c.2041G>A (p.Ala681Thr), which has been reported as pathogenic despite inconclusive data on protein expression and normal MMR activity resulting from experimental evaluation of different studies [207].

Our analysis also showed that 18 PDAC patients (20.7%) had PVs/LPVs in the *MSH2* gene. Intriguingly, these PVs/LPVs (missense, splicing, and truncating variants) were found to cluster in a hotspot ranging from nucleotide 900 to nucleotide 1300 of the *MSH2* coding region. The *MSH2* missense variant c.1046C>T (p.Pro349Leu) was found in three patients [208].

Moreover, we found that 26 PDAC patients carried distinct germline PVs/LPVs in the *MSH6* gene. Of these, truncating variants were identified with a higher frequency (22/26, 84.6%) as compared to other alteration types (missense and splicing), which were found in four patients (15.4%).

Concerning *PMS2*, a total of 20 PDAC patients were identified as harboring genetic alterations in this gene. Interestingly, a higher rate of PDAC patients (9/20, 45%) had *PMS2* truncating variants as compared to PDAC patients (6/20, 30%) with *PMS2* missense variants.

MSI-H pancreatic cancers are defective in DNA MMR and can occur in inherited disorders such as LS. Patients with this type of pancreatic cancer are less responsive to fluorouracil and gemcitabine and more responsive to FOLFIRINOX [209,210]. Furthermore, pembrolizumab, an inhibitor of the immune checkpoint protein PD–1, has been approved for several MMR-deficient tumor types [211]. Indeed, a review of immune–based treatment approaches for patients with MSI–H pancreatic cancer suggests that immune checkpoint inhibition therapy is effective and has the potential to provide a good response in this subgroup of patients [212].

Of note, a subset of pancreatic cancers arises in the background of Peutz–Jeghers syndrome (PJS), which is caused by germline alterations in the *STK11* gene. *STK11* encodes for a tumor suppressor serine/threonine protein kinase that controls the activity of AMPK family members [213]. Individuals with PJS have gastrointestinal polyps and an increased risk of CRC as well as extracolonic malignancies, including stomach, lung, breast, ovarian, cervical, testes, and pancreatic cancer [178]. Based on literature data, the cumulative risk of developing PDAC at the age of 70 years in these patients ranges from 11 to 36%, which represents a 132–fold increase compared to the general population [214,215]. Our literature review analysis showed that 10 germline variants in the *STK11* gene were detected in 11 (6.1%) PDAC patients. Most of these alterations (*n* = 7, 70%) were truncating variants, which were interspersed throughout the region encoding for STK11 kinase catalytic domain (aa 49–309). The remaining alterations (*n* = 3, 30%) were splicing variants and large deletions involving the promoter and the first exons (exon 1 and/or 3) of the *STK11* gene. Importantly, somatic alterations in the *STK11* gene have emerged as potential therapy targets in patients with non–small cell lung cancer (NSCLC), an approach that may also lead to novel therapeutic opportunities in PDAC [216]. Germline alterations in the *CDKN2A* (p16INK4a/p14ARF) gene are responsible for familial melanoma–pancreatic cancer syndrome. In addition, *CDKN2A* is a susceptibility gene for head and neck squamous cell carcinoma (HNSCC), lung cancer, esophageal cancer, neural system cancer, breast carcinoma, and sarcomas [217]. *CDKN2A* is a tumor suppressor gene playing a central role in the regulation of the cell cycle and induction of apoptosis. Patients with *CDKN2A* germline alterations have a 20– to 47–fold increased risk of developing PDAC [218,219]. Our literature review analysis showed that 41 germline variants in the *CDKN2A* gene were detected in 146 PDAC patients. Twenty–seven of these alterations were found to be recurrent, i.e., occurring in more than one individual, while the remaining fourteen were identified in one patient each. Several of these *CDKN2A* PVs/PLVs were previously identified as founder mutations in different populations: c.301G>T, p.Gly101Trp (detected in 26 PDAC patients) in southeastern Europe [220], c.377T>A, p.Val126Asp (detected in 13 PDAC patients) in North America [221], c.335_337dup, p.Arg112dup (detected in 11 PDAC patients) in Sweden [222], and c.225_243del, p.Ala76Cysfs*64 (detected in 7 PDAC patients) in the Netherlands [223]. Moreover, another *CDKN2A* founder mutation (c.159G>C; p.Met53Ile), mostly occurring in the Scottish population [224], was detected in eight patients. While limited achievements have been reported in targeted monotherapy in pancreatic cancer harboring *CDKN2A* loss-of-function genomic alterations [225], a recent pilot study demonstrated efficacy in a small group of PDAC patients with *CDKN2A* somatic alterations treated with a combination of genomically matched targeted agents [226].

Germline mutations in the *TP53* gene are responsible for Li–Fraumeni syndrome, which is characterized by autosomal dominant inheritance and early onset of a wide range of tumor types, including soft tissue sarcomas and osteosarcomas, breast cancer, brain tumors, leukemia, and adrenocortical carcinoma. Moreover, this syndrome is characterized by a high risk of pancreatic cancer [181]. Our literature review analysis showed that a few germline variants in the *TP53* gene were detected in 24 (13.2%) PDAC patients, most of which were missense mutations (75%) interspersed throughout the coding region of the gene. Moreover, three missense variants (c.542G>A, p.Arg181His; c.736A>G, p.Met246Val; c.844C>T, p.Arg282Trp) were identified in each of two unrelated individuals. Of note, *TP53* mutations are critical drivers that influence the carcinogenesis and prognosis of PDAC patients [227]. Recently, a phase I trial demonstrated the therapeutic efficacy of gene therapy based on SGT–53, an antitumor agent comprising a cationic liposome encapsulating a plasmid encoding wild–type p53, in the treatment of different solid tumors [228]. Interestingly, a phase II clinical trial of SGT–53 plus gemcitabine and nab–paclitaxel (NCT02340117) is being conducted for advanced PDAC. Overall, the mutational status of the *TP53* gene may prove useful to guide therapeutic strategies in PDAC patients [229].

In the era of precision medicine, the integration of germline and somatic genetic profiling is gaining a central role in medical oncology. Indeed, germline and somatic genetic testing may help personalize therapeutic decisions for targeted therapies according to each patient’s mutational status [230]. Moreover, germline characterization of genetic variants underlying cancer susceptibility may enable the identification of individuals at risk of developing a particular cancer and guide precision prevention strategies [231]. Each individual’s overall risk of cancer depends at least in part on the type of genetic variant and the phenotypic effect of the altered genes. As regards PDAC susceptibility genes, *BRCA1* and *BRCA2* are considered high penetrance genes in breast and ovarian cancer, and *MLH1* and *MSH2* are considered high penetrance genes in LS CRC [231]. On the other hand, *ATM* and *PALB2* are considered moderate penetrance genes in breast and ovarian cancer [232], and *MSH6*, *PMS2*, and *EPCAM* are considered moderate penetrance genes in LS CRC [233]. Additionally, *CDKN2A*, *STK11*, and *TP53* are considered highly penetrant genes in melanoma, juvenile polyposis syndrome (JPS), and Li–Fraumeni syndrome, respectively. In this study, a high proportion of PDAC patients were found to harbor truncating variants, which are expected to have a more deleterious effect on gene function, in moderate penetrance genes (*ATM*, *PALB2*, *MSH6*, *PMS2*). Conversely, alterations that are likely to have a lesser impact on gene function, such as missense variants, were predominantly detected in the subgroup of high penetrance genes (*CDKN2A, TP53*) in these patients. Instead, missense and truncating variants involving the *MLH1* and *MSH2* genes were identified in a similar proportion of PDAC patients. As regards the *STK11* gene, which has high penetrance for the development of mucocutaneous pigmentation and gastrointestinal hamartomatous polyposis in JPS, a higher proportion of PDAC patients were found to carry truncating variants as compared to missense variants. Moreover, truncating variants were predominantly detected in the high penetrant genes *BRCA1* and *BRCA2* in these patients.

Altogether, these findings help decipher the genetic landscape of relevant cancer susceptibility genes in PDAC patients. As such, they may support the development of tailored management and follow–up strategies in PDAC patients with specific germline genetic variants and a personal and/or familial history of cancer.

However, this systematic review study has various limitations.

First, we only included studies that focused on PDAC cancer patients who underwent genetic testing for hereditary syndromes known to be associated with an increased risk of PDAC, which limits a more comprehensive understanding of the role of other emerging genes on genetic predisposition to PDAC. Second, the present systematic literature review includes studies published from 1976 to 2023. The inclusion of outdated scientific articles may have influenced the accuracy or relevance of the results due to older methodologies associated with clinical screening and genetic testing. Third, this systematic review study excludes scientific articles not providing a detailed genetic annotation of genetic variants, which limits a broader integration of clinical information of other PDAC patients. Moreover, the present study includes PDAC patients described in scientific articles, which may not reflect the broader global diversity of all patients with PDAC, thus weakening the generalizability of the findings. A further limitation of the present study is represented by the inclusion of PDAC patients with a known genetic predisposition to PDAC. Notably, PDAC is a multifactorial disease influenced by a combination of genetic, environmental, and lifestyle factors. The integration of these aspects into future research could provide a more holistic understanding of PDAC risk and development.

This systematic review study aimed to decipher the genetic landscape of patients with any type or stage of PDAC. In this regard, the lack of standardized clinical data at the individual patient level in the selected articles limits the comprehension of specific associations between genetic findings and disease characteristics.

## 5. Conclusions

In this systematic literature review study, we explored the genetic landscape of PDAC in patients with germline genetic variants in major PDAC susceptibility genes. The molecular characterization of these patients highlights the importance of personalized medicine to provide tailored genetic counseling, management, and surveillance to families with PDAC and hereditary cancer.

## Figures and Tables

**Figure 1 cancers-16-00056-f001:**
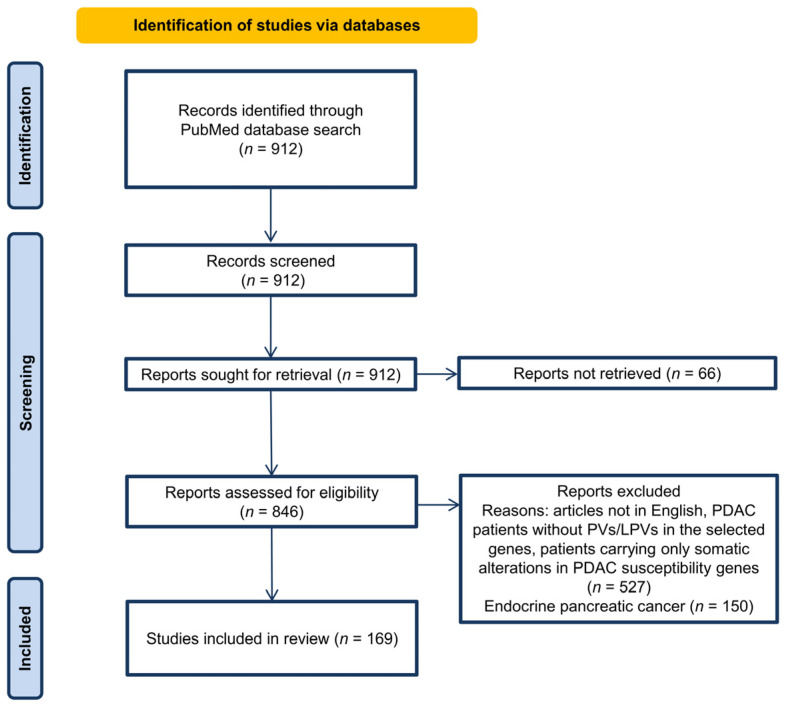
PRISMA flow-chart diagram showing the article selection process.

**Figure 2 cancers-16-00056-f002:**
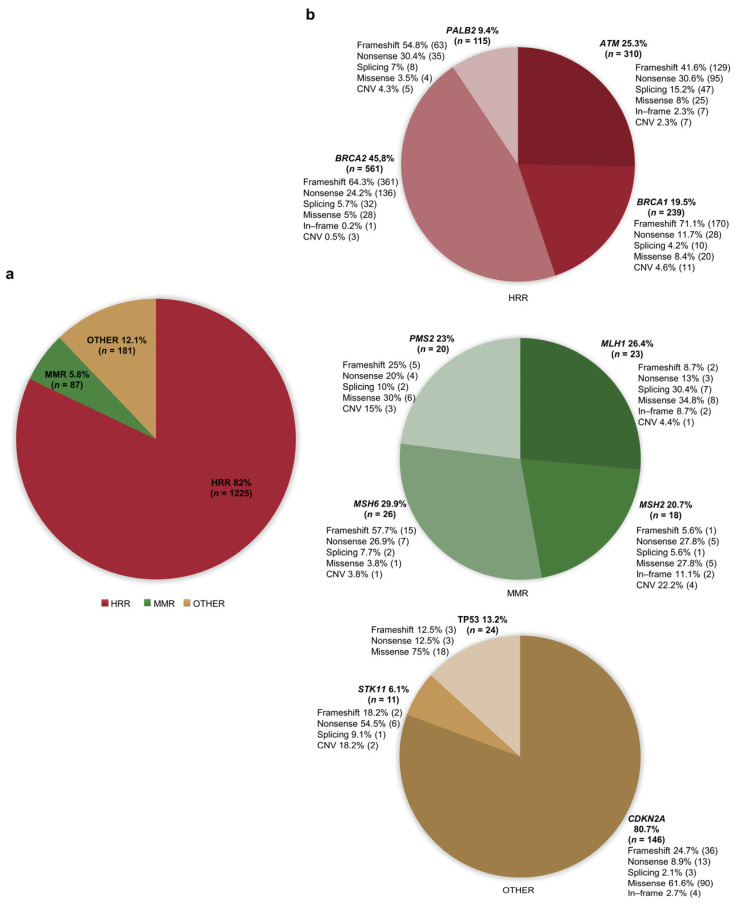
Percentage distribution of patients with pancreatic ductal adenocarcinoma (PDAC) carrying germline pathogenic/likely pathogenic variants (PVs/LPVs). (**a**) Distribution of patients based on functional gene groups (HRR, MMR, and others). (**b**) Distribution of patients with PVs/LPVs in MMR, HRR, and other genes based on the specific gene. Abbreviations: CNV: copy number variation; HRR: homologous recombination repair; MMR: mismatch repair.

**Figure 3 cancers-16-00056-f003:**
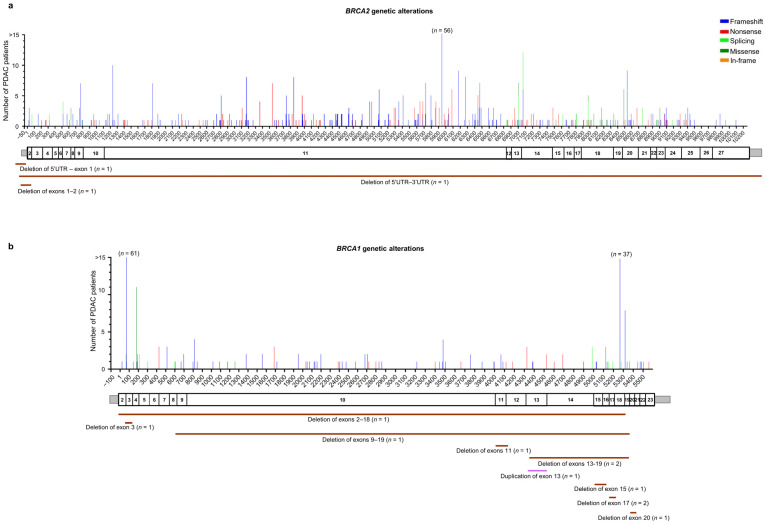
Distribution of the alterations identified in HRR genes in the pancreatic ductal adenocarcinoma (PDAC) patients analyzed in this study: (**a**) *BRCA2* (NM_000059.4), (**b**) *BRCA1* (NM_7294.4), (**c**) *ATM* (NM_000051.4), and (**d**) *PALB2* (NM_024675.4). For each gene, the upper panel shows the identified genetic alterations, with colored vertical bars representing the type of pathogenic variants (PVs) and likely pathogenic variants (LPVs). Color codes are as follows: blue bars = frameshift variants, red bars = nonsense variants, light green bars = splicing variants, dark green bars = missense variants, orange bars = in–frame variants. The height of the bar represents the number of PDAC patients harboring a genetic variant at the specified position. The x-axis represents the coding sequence of each gene. For each gene, the lower panel shows a schematic representation of its exons (rectangles). Copy number variants (CNVs) are represented by color codes: brown lines = large deletions, violet lines = large duplications.

**Figure 4 cancers-16-00056-f004:**
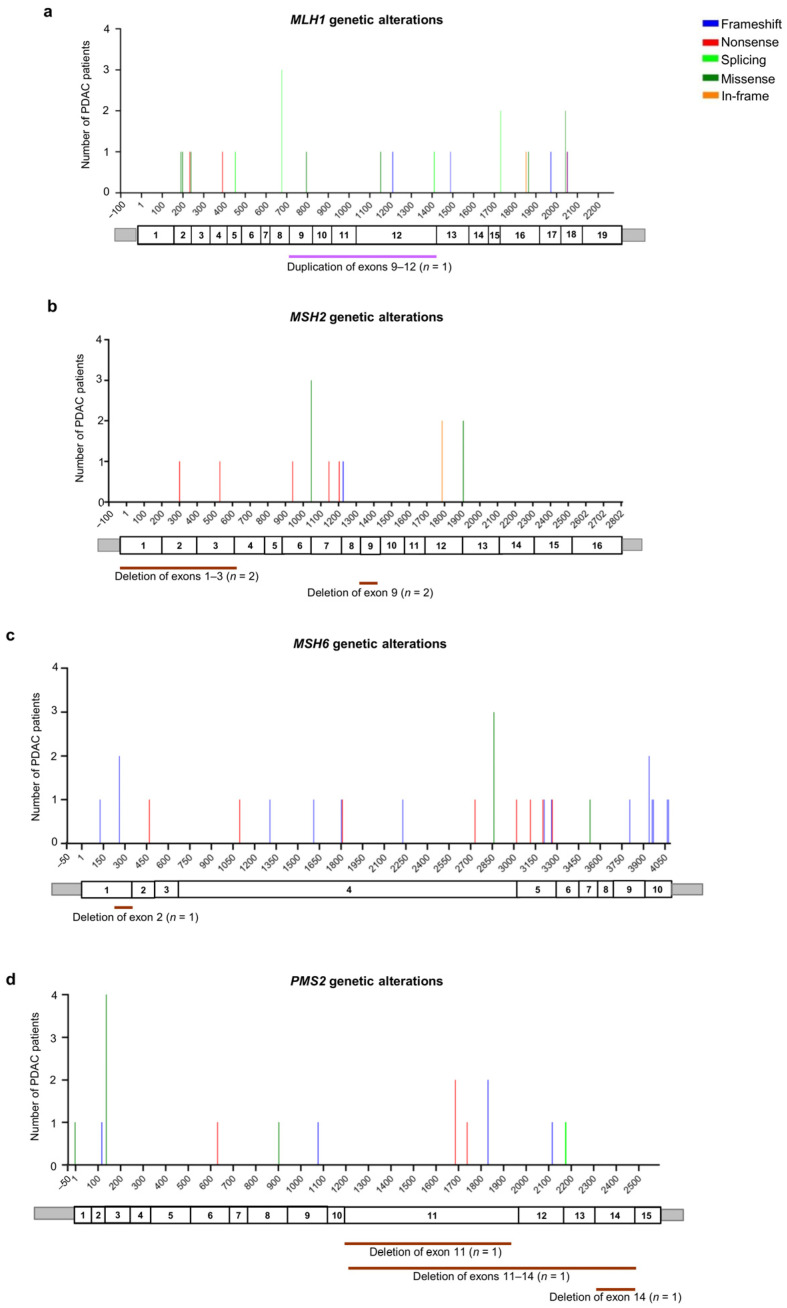
Distribution of the alterations identified in MMR genes in the pancreatic ductal adenocarcinoma (PDAC) patients analyzed in this study: (**a**) *MLH1* (NM_000249.4), (**b**) *MSH2* (NM_000251.3), (**c**) *MSH6* (NM_000179.3), and (**d**) *PMS2* (NM_000535.7). For each gene, the upper panel shows the identified genetic alterations, with colored vertical bars representing the type of pathogenic variants (PVs) and likely pathogenic variants (LPVs). Color codes are as follows: blue bars = frameshift variants, red bars = nonsense variants, light green bars = splicing variants, dark green bars = missense variants, orange bars = in-frame variants. The height of the bar represents the number of PDAC patients harboring a genetic variant at the specified position. The x-axis represents the coding sequence of each gene. For each gene, the lower panel shows a schematic representation of its exons (rectangles). Copy number variants (CNVs) are represented by color codes: brown lines = large deletions, violet lines = large duplications.

**Figure 5 cancers-16-00056-f005:**
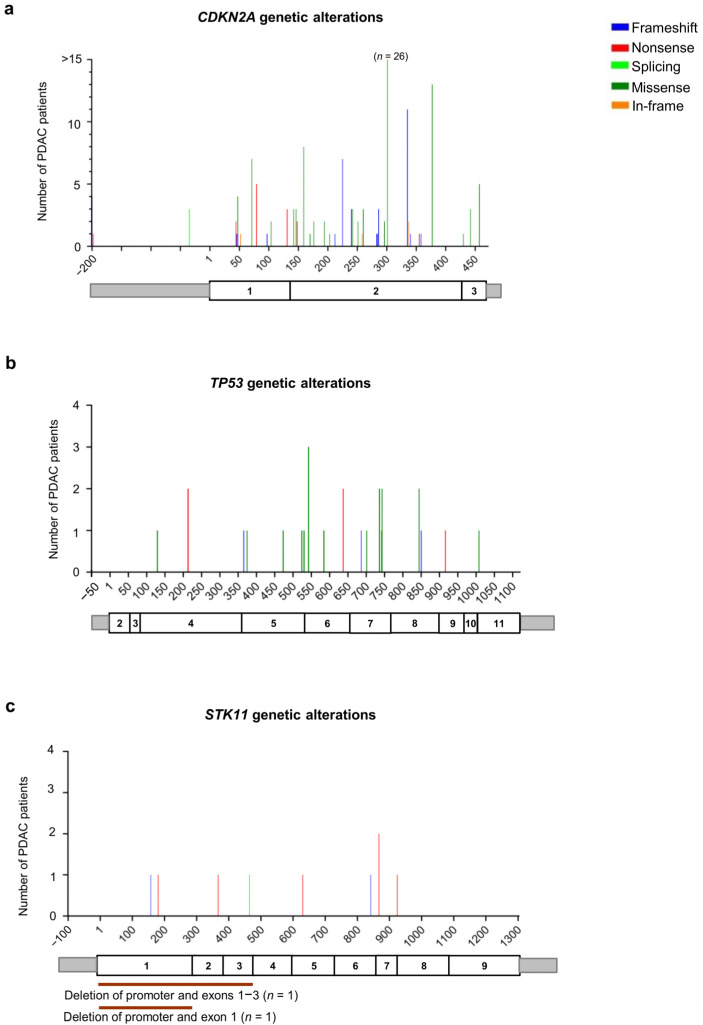
Distribution of the alterations identified in genes involved in other cancer-related pathways in the pancreatic ductal adenocarcinoma (PDAC) patients analyzed in this study: (**a**) *CDKN2A* (NM_000077.5), (**b**) *TP53* (NM_000546.6), and (**c**) *STK11* (NM_000455.5). For each gene, the upper panel shows the identified genetic alterations, with colored vertical bars representing the type of pathogenic variants (PVs) and likely pathogenic variants (LPVs). Color codes are as follows: blue bars = frameshift variants, red bars = nonsense variants, light green bars = splicing variants, dark green bars = missense variants, orange bars = in-frame variants. The height of the bar represents the number of PDAC patients harboring a genetic variant at the specified position. The x-axis represents the coding sequence of each gene. For each gene, the lower panel shows a schematic representation of its exons (rectangles). Copy number variants (CNVs) are represented by color codes: brown lines = large deletions, violet lines = large duplications.

## Data Availability

The data presented in this study are available in this article.

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
