# Peer review of "Understanding the Genetic Landscape of Pancreatic Ductal Adenocarcinoma to Support Personalized Medicine: A Systematic Review"

_cancers, 2023, doi:10.3390/cancers16010056_

Round 1

Reviewer 1 Report

Comments and Suggestions for Authors

I found this paper by Pantaleo et al interesting and clear. The topic is of clinical relevance with respect to the current management of families with PDAC. The authors included the relevant literature and nicely analyzed and exposed the data. I have only minor points to be changed/discussed.

 Material and Methods

1-    The timeframe of the search was from February 1976 to December 2023: please correct with October or November 2023. It is methodologically incorrect to say You have searched until December 2023.

 Results

1-    In the first sub-heading it would be nice to add the data regarding the study design of the paper included in the analysis.

2-    I would suggest adding a Table with general features of the included papers (i.e., author, year, study design, population, general mutation details).

3-    Do you have any data/comment regarding the indications of the patients included in the different studies to perform genetic testing?

4-    Do You have any additional clinical data regarding the patients included in the different studies (i.e., age < 50, metachronous associated/non-associated neoplasms)?

Discussion

This section is very nicely exposed, however I would recommend to shorten it a little bit. Some redundant data ca be cut.

1-    It can be interesting to add a few sentences/a small paragraph regarding the surveillance in high-risk patients including individuals with PVs (see. Paiella S, Capurso G, Carrara S, et al. Outcomes of a 3-year prospective surveillance in individuals at high-risk for pancreatic cancer. Am J Gastroenterol. 2023 Oct 3. doi: 10.14309/ajg.0000000000002546).

 Overall, I would suggest improving the definition/quality of the Figures.

Comments on the Quality of English Language

Minor revision of English is required.

Author Response

Dear Cancers Editorial Team,

we are pleased to submit the amended version of our work “Understanding the genetic landscape of pancreatic ductal adenocarcinoma to support personalized medicine: a systematic review” (Cancers-2737572), which we would like to have considered for publication in Cancers journal. We addressed below all the comments raised by the Reviewers, mainly by responding to their observations, clarifying important points, and making the suggested corrections in the text. Moreover, we improved the resolution/quality of all the figures in the manuscript. In addition, we added a new Supplementary Table 1 (Table S1), which includes the PRISMA checklist according to Cancers journal guidelines, and a new Supplementary Table 2 (Table S2), which summarizes the data characteristics of the included studies. As a result, the order of the Supplementary Tables has changed, and the Supplementary Table 1 included in the previous version of the manuscript is now Supplementary Table 3 (Table S3).

Reviewer 1

Open Review

Quality of English Language

( ) I am not qualified to assess the quality of English in this paper
( ) English very difficult to understand/incomprehensible
( ) Extensive editing of English language required
( ) Moderate editing of English language required
(x) Minor editing of English language required
( ) English language fine. No issues detected

Yes

Can be improved

Must be improved

Not applicable

Does the introduction provide sufficient background and include all relevant references?

(x)

( )

( )

( )

Are all the cited references relevant to the research?

(x)

( )

( )

( )

Is the research design appropriate?

(x)

( )

( )

( )

Are the methods adequately described?

( )

(x)

( )

( )

Are the results clearly presented?

( )

(x)

( )

( )

Are the conclusions supported by the results?

( )

(x)

( )

( )

Comments and Suggestions for Authors

I found this paper by Pantaleo et al interesting and clear. The topic is of clinical relevance with respect to the current management of families with PDAC. The authors included the relevant literature and nicely analyzed and exposed the data. I have only minor points to be changed/discussed.

 Material and Methods

  • The timeframe of the search was from February 1976 to December 2023: please correct with October or November 2023. It is methodologically incorrect to say You have searched until December 2023.

We thank the reviewer for this comment. In the previous version of the manuscript, we erroneously stated the following: “The timeframe of the search was from February 1976 to December 2023”. The timeframe of the search was actually from February 1976 to September 2023, thus in this amended version of the manuscript, we corrected this information by replacing “December 2023” with “September 2023”.

Results

  • In the first sub-heading it would be nice to add the data regarding the study design of the paper included in the analysis.

We thank the reviewer for this comment. As suggested, in this amended version of the manuscript, we added summary information regarding the study design of the scientific articles included in this systematic review study. Specifically, in the Results section (paragraph 3.1 “Literature search and study selection”), we added the following sentence “The selected 170 published studies included different types of study designs, mostly consisting of case studies and cohort studies.” Moreover, we summarized the general features of the selected papers (including the study design) in the new Supplementary Table 2 (Table S2, worksheet “Data characteristics of the included studies”).

2-    I would suggest adding a Table with general features of the included papers (i.e., author, year, study design, population, general mutation details).

We thank the reviewer for this comment. As indicated above, in this amended version of the manuscript, we summarized the general features of the selected papers (i.e., author, year, study design, population, general mutation details) in the new Supplementary Table 2 (Table S2, worksheet “Data characteristics of the included studies”). In order to address the comments from other reviewers, we also added a new Supplementary Table 1 (Table S1), which includes the PRISMA checklist according to Cancers journal guidelines. As a result, the order of the Supplementary Tables has changed, and the Supplementary Table 1 included in the previous version of the manuscript is now Supplementary Table 3 (Table S3).

3-    Do you have any data/comment regarding the indications of the patients included in the different studies to perform genetic testing?

We thank the reviewer for this comment. As suggested, in this amended version of the manuscript, we added the indication of the patients included in the different studies to perform genetic analyses. Specifically, in the Results section (paragraph “3.2. Gene alteration frequency in PDAC patients”), we added the following sentence: “Genetic testing was performed in PDAC individuals with or without a family history of different cancer types (i.e., pancreatic, breast, colorectal cancer, or melanoma).”

4-    Do You have any additional clinical data regarding the patients included in the different studies (i.e., age < 50, metachronous associated/non-associated neoplasms)?

We thank the reviewer for this useful observation. We agree with the reviewer that it would be very interesting to recognize the potential associations between germline genetic variants and PDAC patients’ clinical data (i.e., age < or > 50 years, metachronous associated/non-associated neoplasms, stage of disease, survival). Indeed, future studies could benefit from stratifying patients based on genetic and clinical data to provide tailored clinical surveillance for high-risk individuals. Unfortunately, we were not able to extract consistent clinical data from the papers included in this systematic literature review due to the following: i) not all of the papers included clinical information; ii) when included, the clinical data were not reported in a homogeneous, comparable way in the different articles.

Discussion

This section is very nicely exposed, however I would recommend to shorten it a little bit. Some redundant data ca be cut.

  • It can be interesting to add a few sentences/a small paragraph regarding the surveillance in high-risk patients including individuals with PVs (see. Paiella S, Capurso G, Carrara S, et al. Outcomes of a 3-year prospective surveillance in individuals at high-risk for pancreatic cancer. Am J Gastroenterol. 2023 Oct 3. doi: 10.14309/ajg.0000000000002546).

We thank the reviewer for this observation. As suggested, in this amended version of the manuscript, we reorganized the Discussion section in order to remove redundancies. Furthermore, in the initial part of the Discussion section, we highlighted the importance of clinical surveillance programs for high-risk patients. 

Overall, I would suggest improving the definition/quality of the Figures.

We thank the reviewer for this recommendation. As suggested, in this amended version of the manuscript, we improved the definition/quality of the Figures.

Comments on the Quality of English Language

Minor revision of English is required.

We thank the Reviewer for this advice. We carefully revised the manuscript to improve grammar and readability.

Submission Date 09 November 2023 Date of this review 10 Nov 2023 09:29:22

Reviewer 2 Report

Comments and Suggestions for Authors

The manuscript entitled “Understanding the genetic landscape of pancreatic ductal adenocarcinoma to support personalized medicine: a systematic review” is a meta-analysis demonstrating the prevalence of germline mutations in pancreatic ductal adenocarcinomas in genes involved in DNA repair machinery including homologous recombination. Overall, this study is interesting and focuses on an understudied with translational impact and is expected to be of interest both to basic scientists as well as clinicians. Below there are two comments which could strengthen the manuscript.

First, could the authors list the pathogenic mutations found in the genes analyzed.

Second, could the authors associate the different mutations with clinicopathological data (i.e. TNM stage and survival).

Author Response

Dear Cancers Editorial Team,

we are pleased to submit the amended version of our work “Understanding the genetic landscape of pancreatic ductal adenocarcinoma to support personalized medicine: a systematic review” (Cancers-2737572), which we would like to have considered for publication in Cancers journal. We addressed below all the comments raised by the Reviewers, mainly by responding to their observations, clarifying important points, and making the suggested corrections in the text. Moreover, we improved the resolution/quality of all the figures in the manuscript. In addition, we added a new Supplementary Table 1 (Table S1), which includes the PRISMA checklist according to Cancers journal guidelines, and a new Supplementary Table 2 (Table S2), which summarizes the data characteristics of the included studies. As a result, the order of the Supplementary Tables has changed, and the Supplementary Table 1 included in the previous version of the manuscript is now Supplementary Table 3 (Table S3).

Reviewer 2

Open Review

Quality of English Language

( ) I am not qualified to assess the quality of English in this paper
( ) English very difficult to understand/incomprehensible
( ) Extensive editing of English language required
( ) Moderate editing of English language required
( ) Minor editing of English language required
(x) English language fine. No issues detected

Yes

Can be improved

Must be improved

Not applicable

Does the introduction provide sufficient background and include all relevant references?

(x)

( )

( )

( )

Are all the cited references relevant to the research?

(x)

( )

( )

( )

Is the research design appropriate?

(x)

( )

( )

( )

(x)

( )

( )

( )

Are the results clearly presented?

(x)

( )

( )

( )

Are the conclusions supported by the results?

( )

( )

( )

( )

Comments and Suggestions for Authors

The manuscript entitled “Understanding the genetic landscape of pancreatic ductal adenocarcinoma to support personalized medicine: a systematic review” is a meta-analysis demonstrating the prevalence of germline mutations in pancreatic ductal adenocarcinomas in genes involved in DNA repair machinery including homologous recombination. Overall, this study is interesting and focuses on an understudied with translational impact and is expected to be of interest both to basic scientists as well as clinicians. Below there are two comments which could strengthen the manuscript.

First, could the authors list the pathogenic mutations found in the genes analyzed.

We thank the reviewer for this suggestion. In this amended version of the manuscript, we added information about the clinical significance of germline variants. Specifically, in each worksheet of the amended version of Supplementary Table S3 (Table S3), which lists the germline variants identified for each of the selected genes (ATM, BRCA1, BRCA2, CDKN2A, EPCAM, MLH1, MSH2, MSH6, PALB2, PMS2, STK11, and TP53), we added a new column entitled “Variant classification” to include the clinical classification (“pathogenic” or “likely pathogenic”) of each variant. In order to address the comments from other reviewers, we also added a new Supplementary Table 1 (Table S1), which includes the PRISMA checklist according to Cancers journal guidelines, and a new Supplementary Table 2 (Table S2), which summarizes the data characteristics of the included studies. As a result, the order of the Supplementary Tables has changed, and the Supplementary Table 1 included in the previous version of the manuscript is now Supplementary Table 3 (Table S3).

Second, could the authors associate the different mutations with clinicopathological data (i.e. TNM stage and survival).

We thank the reviewer for this useful observation. We agree with the reviewer that it would be very interesting to recognize the potential associations between germline genetic variants and PDAC patients’ clinical data (i.e., age < or > 50 years, metachronous associated/non-associated neoplasms, stage of disease, survival). Indeed, future studies could benefit from stratifying patients based on genetic and clinical data to provide tailored clinical surveillance for high-risk individuals. Unfortunately, we were not able to extract consistent clinical data from the papers included in this systematic literature review due to the following: i) not all of the papers included clinical information; ii) when included, the clinical data were not reported in a homogeneous, comparable way in the different articles.

Submission Date 09 November 2023 Date of this review 30 Nov 2023 15:19:13

Reviewer 3 Report

Comments and Suggestions for Authors

The authors of this study focus on Pancreatic Ductal Adenocarcinoma (PDAC), a highly aggressive cancer known for its late-stage presentation and high mortality rate. They emphasize the critical need for early detection, particularly in individuals at high risk of PDAC. The study advocates for the participation of these high-risk individuals in clinical management and surveillance programs. Specifically, these programs are recommended for individuals with a germline pathogenic variant in a cancer predisposition gene or those with a strong family history of PDAC.

To deepen understanding of PDAC susceptibility, the authors conducted a systematic literature review analyzing the mutation patterns in key genes associated with PDAC risk, such as ATM, BRCA1, BRCA2, CDKN2A, EPCAM, MLH1, MSH2, MSH6, PALB2, PMS2, STK11, and TP53. They examined 1493 PDAC patients from various studies and found that a significant majority (82%) had alterations in genes linked to the homologous recombination repair (HRR) pathway (ATM, BRCA1, BRCA2, PALB2). A smaller proportion had mutations in genes associated with other cancer pathways (CDKN2A, STK11, TP53) or the mismatch repair (MMR) pathway (MLH1, MSH2, MSH6, PMS2).

The findings underscore the importance of genetic profiling in PDAC patients. Identifying germline genetic variants can support the development of personalized treatment approaches, improve clinical management, and enhance surveillance strategies for patients with PDAC. This targeted approach based on genetic characterization could potentially lead to more effective treatments and better outcomes for PDAC patients.

  1. Scope of Genetic Variants Studied: The study focuses on specific genes known to be associated with PDAC susceptibility. However, this approach may overlook other genetic factors that could play a role in PDAC development. Expanding the scope to include emerging or less-studied genes could provide a more comprehensive understanding of genetic predispositions to PDAC.

  2. Timeframe of Literature Review: The literature review covers publications from 1976 to 2023. While this extensive range includes a wealth of historical data, it might also mean that older studies with outdated methodologies or less advanced genetic testing capabilities are included. This could potentially affect the accuracy or relevance of the findings. A focus on more recent studies might yield more accurate and applicable results.

  3. Population Heterogeneity: The study includes individuals with any type or stage of PDAC, which could introduce significant heterogeneity into the data. This wide net might obscure specific genetic associations that are more pronounced in certain subtypes or stages of the disease. Future studies could benefit from stratifying patients based on disease characteristics to identify more precise genetic correlations.

  4. Exclusion Criteria: The exclusion of studies not providing detailed genetic nomenclature is prudent, but it might also exclude valuable clinical data. It's essential to balance the need for genetic specificity with the potential insights that broader clinical data could provide. Future research could consider integrating broader clinical outcomes to complement genetic findings.

  5. Reviewer Bias in Data Extraction: Although the data extraction was performed by ten independent reviewers, there is still potential for subjective interpretation, especially in cases where there was disagreement. To mitigate this, implementing more stringent criteria for study inclusion and a more structured consensus process for resolving disagreements could be beneficial.

  6. Generalizability of Results: The study's results are based on a systematic literature review, which might limit the generalizability of the findings. The populations studied in the reviewed literature may not reflect the broader global diversity, and therefore, the results may not be applicable to all PDAC patients. Future studies could aim to include a more diverse patient population.

  7. Consideration of Environmental and Lifestyle Factors: The study primarily focuses on genetic predispositions to PDAC. However, PDAC is a complex disease likely influenced by a combination of genetic, environmental, and lifestyle factors. Integrating these aspects into future research could provide a more holistic understanding of PDAC risk and development.

In conclusion, while the study provides valuable insights into the genetic landscape of PDAC, addressing these limitations and considering these suggestions in future research could enhance our understanding of PDAC genetics and improve clinical management and surveillance strategies for high-risk individuals.

Next,

Bone metastasis as the primary presentation of pancreatic ductal adenocarcinoma (PDAC) is a relatively rare but significant clinical phenomenon. PDAC commonly metastasizes to the liver or peritoneum, and bone metastases are less frequent. However, when they occur, they can be the first sign of underlying pancreatic cancer. This situation can be challenging for several reasons:

  1. Diagnostic Challenges: Bone metastasis as the initial manifestation of PDAC can lead to diagnostic delays. Since PDAC typically presents with non-specific symptoms or gastrointestinal signs, bone pain or fractures might not immediately prompt healthcare providers to consider pancreatic cancer.

  2. Prognostic Implications: The occurrence of bone metastasis in PDAC is usually an indicator of advanced disease and poor prognosis. It reflects a more aggressive tumor biology and indicates that the cancer has progressed beyond the pancreas.

  3. Therapeutic Challenges: Treating PDAC with bone metastases involves managing not only the primary tumor but also the skeletal-related events. This may require a multidisciplinary approach involving oncologists, radiologists, orthopedic specialists, and palliative care teams.

  4. Impact on Survival and Quality of Life: Bone metastases can significantly impact a patient's quality of life due to pain, mobility issues, and the risk of pathological fractures. These complications can also affect overall survival rates.

  5. Research Implications: The phenomenon underscores the need for further research into the mechanisms of metastasis in PDAC. Understanding why and how pancreatic cancer cells metastasize to bone could open new avenues for treatment and prevention.

  6. Need for Early Detection: This scenario highlights the importance of early detection strategies for PDAC. Given its aggressive nature and propensity for early metastasis, improved screening and surveillance techniques could be crucial in identifying the disease before it progresses to an advanced stage. please refer to and expand according to doi.org/10.1002/ccr3.2412

Comments on the Quality of English Language

The authors of this study focus on Pancreatic Ductal Adenocarcinoma (PDAC), a highly aggressive cancer known for its late-stage presentation and high mortality rate. They emphasize the critical need for early detection, particularly in individuals at high risk of PDAC. The study advocates for the participation of these high-risk individuals in clinical management and surveillance programs. Specifically, these programs are recommended for individuals with a germline pathogenic variant in a cancer predisposition gene or those with a strong family history of PDAC.

To deepen understanding of PDAC susceptibility, the authors conducted a systematic literature review analyzing the mutation patterns in key genes associated with PDAC risk, such as ATM, BRCA1, BRCA2, CDKN2A, EPCAM, MLH1, MSH2, MSH6, PALB2, PMS2, STK11, and TP53. They examined 1493 PDAC patients from various studies and found that a significant majority (82%) had alterations in genes linked to the homologous recombination repair (HRR) pathway (ATM, BRCA1, BRCA2, PALB2). A smaller proportion had mutations in genes associated with other cancer pathways (CDKN2A, STK11, TP53) or the mismatch repair (MMR) pathway (MLH1, MSH2, MSH6, PMS2).

The findings underscore the importance of genetic profiling in PDAC patients. Identifying germline genetic variants can support the development of personalized treatment approaches, improve clinical management, and enhance surveillance strategies for patients with PDAC. This targeted approach based on genetic characterization could potentially lead to more effective treatments and better outcomes for PDAC patients.

Author Response

Dear Cancers Editorial Team,

we are pleased to submit the amended version of our work “Understanding the genetic landscape of pancreatic ductal adenocarcinoma to support personalized medicine: a systematic review” (Cancers-2737572), which we would like to have considered for publication in Cancers journal. We addressed below all the comments raised by the Reviewers, mainly by responding to their observations, clarifying important points, and making the suggested corrections in the text. Moreover, we improved the resolution/quality of all the figures in the manuscript. In addition, we added a new Supplementary Table 1 (Table S1), which includes the PRISMA checklist according to Cancers journal guidelines, and a new Supplementary Table 2 (Table S2), which summarizes the data characteristics of the included studies. As a result, the order of the Supplementary Tables has changed, and the Supplementary Table 1 included in the previous version of the manuscript is now Supplementary Table 3 (Table S3).

Reviewer 3

Open Review

Quality of English Language

( ) I am not qualified to assess the quality of English in this paper
( ) English very difficult to understand/incomprehensible
( ) Extensive editing of English language required
( ) Moderate editing of English language required
(x) Minor editing of English language required
( ) English language fine. No issues detected

Yes

Can be improved

Must be improved

Not applicable

( )

( )

(x)

( )

Are all the cited references relevant to the research?

( )

( )

(x)

( )

Is the research design appropriate?

( )

(x)

( )

( )

Are the methods adequately described?

( )

(x)

( )

( )

Are the results clearly presented?

( )

(x)

( )

( )

Are the conclusions supported by the results?

( )

(x)

( )

( )

Comments and Suggestions for Authors

The authors of this study focus on Pancreatic Ductal Adenocarcinoma (PDAC), a highly aggressive cancer known for its late-stage presentation and high mortality rate. They emphasize the critical need for early detection, particularly in individuals at high risk of PDAC. The study advocates for the participation of these high-risk individuals in clinical management and surveillance programs. Specifically, these programs are recommended for individuals with a germline pathogenic variant in a cancer predisposition gene or those with a strong family history of PDAC.

To deepen understanding of PDAC susceptibility, the authors conducted a systematic literature review analyzing the mutation patterns in key genes associated with PDAC risk, such as ATM, BRCA1, BRCA2, CDKN2A, EPCAM, MLH1, MSH2, MSH6, PALB2, PMS2, STK11, and TP53. They examined 1493 PDAC patients from various studies and found that a significant majority (82%) had alterations in genes linked to the homologous recombination repair (HRR) pathway (ATM, BRCA1, BRCA2, PALB2). A smaller proportion had mutations in genes associated with other cancer pathways (CDKN2A, STK11, TP53) or the mismatch repair (MMR) pathway (MLH1, MSH2, MSH6, PMS2).

The findings underscore the importance of genetic profiling in PDAC patients. Identifying germline genetic variants can support the development of personalized treatment approaches, improve clinical management, and enhance surveillance strategies for patients with PDAC. This targeted approach based on genetic characterization could potentially lead to more effective treatments and better outcomes for PDAC patients.

  1. Scope of Genetic Variants Studied: The study focuses on specific genes known to be associated with PDAC susceptibility. However, this approach may overlook other genetic factors that could play a role in PDAC development. Expanding the scope to include emerging or less-studied genes could provide a more comprehensive understanding of genetic predispositions to PDAC.
  2. Timeframe of Literature Review: The literature review covers publications from 1976 to 2023. While this extensive range includes a wealth of historical data, it might also mean that older studies with outdated methodologies or less advanced genetic testing capabilities are included. This could potentially affect the accuracy or relevance of the findings. A focus on more recent studies might yield more accurate and applicable results.
  3. Population Heterogeneity: The study includes individuals with any type or stage of PDAC, which could introduce significant heterogeneity into the data. This wide net might obscure specific genetic associations that are more pronounced in certain subtypes or stages of the disease. Future studies could benefit from stratifying patients based on disease characteristics to identify more precise genetic correlations.
  4. Exclusion Criteria: The exclusion of studies not providing detailed genetic nomenclature is prudent, but it might also exclude valuable clinical data. It's essential to balance the need for genetic specificity with the potential insights that broader clinical data could provide. Future research could consider integrating broader clinical outcomes to complement genetic findings.
  5. Reviewer Bias in Data Extraction: Although the data extraction was performed by ten independent reviewers, there is still potential for subjective interpretation, especially in cases where there was disagreement. To mitigate this, implementing more stringent criteria for study inclusion and a more structured consensus process for resolving disagreements could be beneficial.
  6. Generalizability of Results: The study's results are based on a systematic literature review, which might limit the generalizability of the findings. The populations studied in the reviewed literature may not reflect the broader global diversity, and therefore, the results may not be applicable to all PDAC patients. Future studies could aim to include a more diverse patient population.
  7. Consideration of Environmental and Lifestyle Factors: The study primarily focuses on genetic predispositions to PDAC. However, PDAC is a complex disease likely influenced by a combination of genetic, environmental, and lifestyle factors. Integrating these aspects into future research could provide a more holistic understanding of PDAC risk and development.

In conclusion, while the study provides valuable insights into the genetic landscape of PDAC, addressing these limitations and considering these suggestions in future research could enhance our understanding of PDAC genetics and improve clinical management and surveillance strategies for high-risk individuals.

We thank the reviewer for these insightful comments. As suggested, in this amended version of the manuscript, we addressed the suggested limitations (points 1, 2, 3, 4, 6, 7) at the end of the Discussion section.

As regards point 5, in the previous version of the manuscript, the following paragraph in the Materials and methods section described the data extraction process

“Ten reviewers independently assessed the title, abstract, main text, and supplementary material of the identified articles to determine study inclusion or exclusion. Relevant information regarding PDAC patients with disease-causative genetic sequence variants was extracted from the full text of the included articles. Any disagreement was resolved by discussion among the reviewers. Extracted study details included information about the article (author(s), year of publication, title, DOI), genetic sequence variant nomenclature (gene, gene transcript, DNA change, protein change, type of variant), and number of patients with the genetic variant.”

We erroneously reported the term “discussion” to indicate the process through which we resolved disagreements. Specifically, disagreements were resolved through a round table discussion using a consensus structured process. Accordingly, in the Materials and Methods section (paragraph “Data Extraction”) of this amended version of the manuscript, we replaced the term “discussion” with “consensus meeting”.

Next,

Bone metastasis as the primary presentation of pancreatic ductal adenocarcinoma (PDAC) is a relatively rare but significant clinical phenomenon. PDAC commonly metastasizes to the liver or peritoneum, and bone metastases are less frequent. However, when they occur, they can be the first sign of underlying pancreatic cancer. This situation can be challenging for several reasons:

  1. Diagnostic Challenges: Bone metastasis as the initial manifestation of PDAC can lead to diagnostic delays. Since PDAC typically presents with non-specific symptoms or gastrointestinal signs, bone pain or fractures might not immediately prompt healthcare providers to consider pancreatic cancer.
  2. Prognostic Implications: The occurrence of bone metastasis in PDAC is usually an indicator of advanced disease and poor prognosis. It reflects a more aggressive tumor biology and indicates that the cancer has progressed beyond the pancreas.
  3. Therapeutic Challenges: Treating PDAC with bone metastases involves managing not only the primary tumor but also the skeletal-related events. This may require a multidisciplinary approach involving oncologists, radiologists, orthopedic specialists, and palliative care teams.
  4. Impact on Survival and Quality of Life: Bone metastases can significantly impact a patient's quality of life due to pain, mobility issues, and the risk of pathological fractures. These complications can also affect overall survival rates.
  5. Research Implications: The phenomenon underscores the need for further research into the mechanisms of metastasis in PDAC. Understanding why and how pancreatic cancer cells metastasize to bone could open new avenues for treatment and prevention.
  6. Need for Early Detection: This scenario highlights the importance of early detection strategies for PDAC. Given its aggressive nature and propensity for early metastasis, improved screening and surveillance techniques could be crucial in identifying the disease before it progresses to an advanced stage. please refer to and expand according to org/10.1002/ccr3.2412

We thank the reviewer for these valuable observations. As pointed out by the reviewer, the majority of patients with PDAC present with metastasis at the time of initial diagnosis or develop metastasis afterward. Thus, early detection of PDAC is a major challenge for clinicians.

As suggested by the Reviewer, in this amended version of the manuscript, we commented on the importance of early detection of PDAC. Specifically, we added the following sentences in the Introduction section: “The most common metastatic sites for PDAC are the liver and peritoneum. Less frequently, lung, brain, and bone metastases are also detected in PDAC patients (doi: 10.3389/fonc.2022.759403. PMID: 35223464). Focusing on early diagnosis of metastatic PDAC, with a particular emphasis on unusual symptoms such as bone pain, which might be related to skeletal metastases, is a key priority to extend survival and improve the quality of life of PDAC patients (doi: 10.1002/ccr3.2412. PMID: 31624620).”

Comments on the Quality of English Language

The authors of this study focus on Pancreatic Ductal Adenocarcinoma (PDAC), a highly aggressive cancer known for its late-stage presentation and high mortality rate. They emphasize the critical need for early detection, particularly in individuals at high risk of PDAC. The study advocates for the participation of these high-risk individuals in clinical management and surveillance programs. Specifically, these programs are recommended for individuals with a germline pathogenic variant in a cancer predisposition gene or those with a strong family history of PDAC.

To deepen understanding of PDAC susceptibility, the authors conducted a systematic literature review analyzing the mutation patterns in key genes associated with PDAC risk, such as ATM, BRCA1, BRCA2, CDKN2A, EPCAM, MLH1, MSH2, MSH6, PALB2, PMS2, STK11, and TP53. They examined 1493 PDAC patients from various studies and found that a significant majority (82%) had alterations in genes linked to the homologous recombination repair (HRR) pathway (ATM, BRCA1, BRCA2, PALB2). A smaller proportion had mutations in genes associated with other cancer pathways (CDKN2A, STK11, TP53) or the mismatch repair (MMR) pathway (MLH1, MSH2, MSH6, PMS2).

The findings underscore the importance of genetic profiling in PDAC patients. Identifying germline genetic variants can support the development of personalized treatment approaches, improve clinical management, and enhance surveillance strategies for patients with PDAC. This targeted approach based on genetic characterization could potentially lead to more effective treatments and better outcomes for PDAC patients.

Reviewer 4 Report

Comments and Suggestions for Authors

The authors performed a systematic literature review to investigate the mutational portrait of the main genes involved in PDAC susceptibility. The molecular characterization of these patients highlights the importance of personalized medicine to provide tailored genetic counseling, management, and surveillance to families with PDAC and hereditary cancer. Albeit, I consider these findings to provide new insight into cancer-related fields, I still have some suggestions.
1, Most figures are highly professional; however, the authors should guide the readers to the meaning of the images appropriately; otherwise, it will likely cause misunderstandings. Therefore, I suggest the author consider revising these figures and legends again.

2, The author performed a systematic literature review to investigate the mutational portrait of the main genes (ATM, BRCA1, BRCA2, CDKN2A, EPCAM, MLH1, MSH2, MSH6, PALB2, PMS2, STK11, TP53) involved in PDAC susceptibility. Since the authors gave a general answer on gene mutation, is there any evidence of different roles in cancer phenotypes of above genes? Please perform pertinent bioinformatic analyses and provide examples of studies investigating miRNA alteration or DNA methylation (https://biit.cs.ut.ee/methsurv/) (PMID: 29264942, 34834441, 33437202).

3, Since Connectivity Map (CMap) can be used to discover the mechanism of action of small molecules, functionally annotate genetic variants of disease genes, and inform clinical trials. It would be fascinating if the main genes (ATM, BRCA1, BRCA2, CDKN2A, 25 EPCAM, MLH1, MSH2, MSH6, PALB2, PMS2, STK11, TP53) could be correlated with other clinical databases. Therefore, I suggest the authors can validate their data via CMap, and discuss these methodologies and literature in the manuscript  (PMID: 29195078, 32064155).

4, There are few typo issues for the authors to pay attention to; please also unify the writing of scientific terms. “Italic, capital”? Please double-check superscripts and subscripts for the whole manuscript.

5, The font is too small for the current figures; meanwhile, the resolution of most figures were quite poor, please improve these data for revision.

Comments on the Quality of English Language

Minor editing of English language required

Author Response

Dear Cancers Editorial Team,

we are pleased to submit the amended version of our work “Understanding the genetic landscape of pancreatic ductal adenocarcinoma to support personalized medicine: a systematic review” (Cancers-2737572), which we would like to have considered for publication in Cancers journal. We addressed below all the comments raised by the Reviewers, mainly by responding to their observations, clarifying important points, and making the suggested corrections in the text. Moreover, we improved the resolution/quality of all the figures in the manuscript. In addition, we added a new Supplementary Table 1 (Table S1), which includes the PRISMA checklist according to Cancers journal guidelines, and a new Supplementary Table 2 (Table S2), which summarizes the data characteristics of the included studies. As a result, the order of the Supplementary Tables has changed, and the Supplementary Table 1 included in the previous version of the manuscript is now Supplementary Table 3 (Table S3).

Reviewer 4

Open Review

Quality of English Language

( ) I am not qualified to assess the quality of English in this paper
( ) English very difficult to understand/incomprehensible
( ) Extensive editing of English language required
( ) Moderate editing of English language required
(x) Minor editing of English language required
( ) English language fine. No issues detected

Yes

Can be improved

Must be improved

Not applicable

Does the introduction provide sufficient background and include all relevant references?

(x)

( )

( )

( )

Are all the cited references relevant to the research?

( )

(x)

( )

( )

Is the research design appropriate?

(x)

( )

( )

( )

Are the methods adequately described?

(x)

( )

( )

( )

Are the results clearly presented?

( )

(x)

( )

( )

Are the conclusions supported by the results?

(x)

( )

( )

( )

Comments and Suggestions for Authors

The authors performed a systematic literature review to investigate the mutational portrait of the main genes involved in PDAC susceptibility. The molecular characterization of these patients highlights the importance of personalized medicine to provide tailored genetic counseling, management, and surveillance to families with PDAC and hereditary cancer. Albeit, I consider these findings to provide new insight into cancer-related fields, I still have some suggestions.

1, Most figures are highly professional; however, the authors should guide the readers to the meaning of the images appropriately; otherwise, it will likely cause misunderstandings. Therefore, I suggest the author consider revising these figures and legends again.

We thank the reviewer for this recommendation. As suggested, in this amended version of the manuscript, we improved the definition/quality of the Figures. Moreover, we revised the figure legends in order to better guide the readers to the meaning of the images.

2, The author performed a systematic literature review to investigate the mutational portrait of the main genes (ATM, BRCA1, BRCA2, CDKN2A, EPCAM, MLH1, MSH2, MSH6, PALB2, PMS2, STK11, TP53) involved in PDAC susceptibility. Since the authors gave a general answer on gene mutation, is there any evidence of different roles in cancer phenotypes of above genes? Please perform pertinent bioinformatic analyses and provide examples of studies investigating miRNA alteration or DNA methylation (https://biit.cs.ut.ee/methsurv/) (PMID: 29264942, 34834441, 33437202).

We thank the reviewer for these comments. As reported in the previous version of the manuscript, and as outlined by the Reviewer, the principal aim of this systematic literature review study was to investigate the mutational portrait of the main genes known to be related to PDAC susceptibility. Notably, the main genes (ATM, BRCA1, BRCA2, CDKN2A, EPCAM, MLH1, MSH2, MSH6, PALB2, PMS2, STK11, TP53) involved in PDAC susceptibility are responsible for known hereditary cancers syndromes conferring a high lifetime risk of a wide range of cancer types. As suggested by the Reviewer, in the Discussion section of this amended version of the manuscript, we commented on the cancer lifetime risk associated with germline alterations of each gene responsible for a hereditary cancer syndrome.

Regarding the second point suggested by the reviewer, which proposes a study of miRNA alteration or DNA methylation, we believe that analyzing genetic and epigenetic PDAC somatic alterations affecting the main genes involved in PDAC susceptibility could be interesting. However, we consider that this evaluation is outside of the scope of the present manuscript. Indeed, in the present manuscript, we stated the following: “we performed a systematic literature review to investigate the mutational portrait of the main genes involved in PDAC susceptibility.” The characterization of germline alterations in different genes allows us to decipher their penetrance and expressivity in hereditary cancer syndromes and PDAC susceptibility, which may help provide tailored clinical surveillance in the context of personalized medicine. Moreover, we would like to stress the following points: i) most PDAC patients included in the present systematic literature review were genetically characterized at the germline level but not at the somatic level; ii) not all of the papers included patients’ clinical information; iii) when included, the clinical data were not reported in a homogeneous, comparable way in the different articles. Hence, no association could be made between genetic (germline and somatic) and clinical data based on the information included and discussed in the manuscript. We agree that investigating the implications of somatic genomic alterations in PDAC and studying their association with PDAC patients’ clinical data may be useful for the characterization of prognostic biomarkers, the molecular classification of PDAC patient subgroups, and the development of novel therapeutic approaches. Similar experimental approaches are described in the articles indicated by the reviewer (PMID: 29264942, 34834441, 33437202). However, this investigation would require a bioinformatics analysis of genomic data reported in cancer datasets (e.g., TCGA, Oncomine, etc.) as no such clinical and molecular data are reported for the PDAC patients discussed in our manuscript. For these reasons, we believe that the study proposed by the reviewer is an interesting project that might be considered for future research.

3, Since Connectivity Map (CMap) can be used to discover the mechanism of action of small molecules, functionally annotate genetic variants of disease genes, and inform clinical trials. It would be fascinating if the main genes (ATM, BRCA1, BRCA2, CDKN2A, 25 EPCAM, MLH1, MSH2, MSH6, PALB2, PMS2, STK11, TP53) could be correlated with other clinical databases. Therefore, I suggest the authors can validate their data via CMap, and discuss these methodologies and literature in the manuscript  (PMID: 29195078, 32064155).

4, There are few typo issues for the authors to pay attention to; please also unify the writing of scientific terms. “Italic, capital”? Please double-check superscripts and subscripts for the whole manuscript.

We thank the reviewer for this observation. In this amended version of the manuscript, we corrected the typos and we harmonized the writing of scientific terms, including superscripts and subscripts.

5, The font is too small for the current figures; meanwhile, the resolution of most figures were quite poor, please improve these data for revision.

We thank the reviewer for this recommendation. As suggested, in this amended version of the manuscript, we improved the definition/quality of the Figures.

Comments on the Quality of English Language

Minor editing of English language required

We thank the Reviewer for this advice. We carefully revised the manuscript to improve grammar and readability.

Submission Date

09 November 2023

Date of this review

01 Dec 2023 13:46:56

Reviewer 5 Report

Comments and Suggestions for Authors

I have only a few minor comments on this important paper on an aggressive cancer.

1. It is written that "Moreover, 3(0.5%) patients were found to harbor different copy number variations (CNVs) involving the BRCA2 gene (Figure 3, Table S1)".  However, in that figure, no duplication (violet) is depicted as far as I can see. As CNVs include not only deletions, but also duplications as on line 181, I found some inconsistency between the description and figure 3c.

2. Similarly, in the text, it is written that "we detected 7 truncating variants (2 frameshift and 5 nonsense) in 8 patiants" (line 218), but in the corresponding figure, it appears that there are 6 nonsense mutations.

3. The English is generally excellent except for unclear usages of the comparative degree such as "higher proportion" or "lower frequency" on line 37, 243, 258, 287, 410, 419: it is not clear what is higher or lower than what.

Author Response

Dear Cancers Editorial Team,

we are pleased to submit the amended version of our work “Understanding the genetic landscape of pancreatic ductal adenocarcinoma to support personalized medicine: a systematic review” (Cancers-2737572), which we would like to have considered for publication in Cancers journal. We addressed below all the comments raised by the Reviewers, mainly by responding to their observations, clarifying important points, and making the suggested corrections in the text. Moreover, we improved the resolution/quality of all the figures in the manuscript. In addition, we added a new Supplementary Table 1 (Table S1), which includes the PRISMA checklist according to Cancers journal guidelines, and a new Supplementary Table 2 (Table S2), which summarizes the data characteristics of the included studies. As a result, the order of the Supplementary Tables has changed, and the Supplementary Table 1 included in the previous version of the manuscript is now Supplementary Table 3 (Table S3).

Reviewer 5

Open Review

Quality of English Language

( ) I am not qualified to assess the quality of English in this paper
( ) English very difficult to understand/incomprehensible
( ) Extensive editing of English language required
( ) Moderate editing of English language required
( ) Minor editing of English language required
(x) English language fine. No issues detected

Yes

Can be improved

Must be improved

Not applicable

Does the introduction provide sufficient background and include all relevant references?

(x)

( )

( )

( )

Are all the cited references relevant to the research?

(x)

( )

( )

( )

Is the research design appropriate?

(x)

( )

( )

( )

Are the methods adequately described?

(x)

( )

( )

( )

Are the results clearly presented?

(x)

( )

( )

( )

Are the conclusions supported by the results?

(x)

( )

( )

( )

Comments and Suggestions for Authors

I have only a few minor comments on this important paper on an aggressive cancer.

  1. It is written that "Moreover, 3(0.5%) patients were found to harbor different copy number variations (CNVs) involving the BRCA2 gene (Figure 3, Table S1)".  However, in that figure, no duplication (violet) is depicted as far as I can see. As CNVs include not only deletions, but also duplications as on line 181, I found some inconsistency between the description and figure 3c.

We thank the Reviewer for this comment. In the previous version of the manuscript (Figure 3, Table S1), we stated the following: “Moreover, 3(0.5%) patients were found to harbor different copy number variations (CNVs) involving the BRCA2 gene (Figure 3, Table S1)”. The different copy number variations (CNVs) were referred to three different large deletions involving the BRCA2 gene. To address the inconsistency mentioned by the Reviewer, in this amended version of the manuscript we corrected the sentences as follows: “Moreover, 3 (0.5%) patients were found to harbor different large deletions involving the BRCA2 gene (Figure 3, Table S3).” (Results Section. Paragraph 3.3. Types of variants identified in HRR genes (ATM, BRCA1, BRCA2, PALB2)). Please also note that the order of the Tables has changed. Indeed, in order to address the comments from other reviewers, we also added a new Supplementary Table 1 (Table S1), which includes the PRISMA checklist according to Cancers journal guidelines, and a new Supplementary Table 2 (Table S2), which summarizes the data characteristics of the included studies. Thus, the Supplementary Table 1 included in the previous version of the manuscript is now Supplementary Table 3 (Table S3)..

  1. Similarly, in the text, it is written that "we detected 7 truncating variants (2 frameshift and 5 nonsense) in 8 patiants" (line 218), but in the corresponding figure, it appears that there are 6 nonsense mutations.

We thank the Reviewer for this comment. In the submitted version of the manuscript, we stated the following: “Among the 11 patients with germline PVs/LPVs (n=10) in the STK11 gene, we detected 7 truncating variants (2 frameshift and 5 nonsense) in 8 patients (72.4%), and a unique splicing variant in a single patient (9.1%). Moreover, 2 deletions involving the promoter and part of the coding region (exon1 and exons 1-3) of the STK11 gene were found in 2 patients (18.2%) (Figure 5, Table S1).” The information reported in the main text, Figure 5, and the amended version of Table S1 (now Table S3), are correct and consistent. Indeed, the STK11 nonsense variant (c.876del; p.Tyr292*) was identified in two different PDAC patients, as shown by the red bar corresponding to two patients on the y-axis (frequency).

  1. The English is generally excellent except for unclear usages of the comparative degree such as "higher proportion" or "lower frequency" on line 37, 243, 258, 287, 410, 419: it is not clear what is higher or lower than what.

We thank the Reviewer for this advice. As suggested, in this amended version of the manuscript, we corrected the sentences containing the expressions "higher proportion" or "lower frequency".

Submission Date

09 November 2023

Date of this review

26 Nov 2023 14:29:46

Round 2

Reviewer 3 Report

Comments and Suggestions for Authors

I am satisfied with the revised version.

Comments on the Quality of English Language

I am satisfied with the revised version.